# The Probiotic *Kluyveromyces lactis* JSA 18 Alleviates Obesity and Hyperlipidemia in High-Fat Diet C57BL/6J Mice

**DOI:** 10.3390/foods13071124

**Published:** 2024-04-07

**Authors:** Yingxiang Hong, Guodong Song, Xiaoqian Feng, Jialei Niu, Lu Wang, Caini Yang, Xuegang Luo, Sa Zhou, Wenjian Ma

**Affiliations:** 1Key Laboratory of Industrial Fermentation Microbiology of the Ministry of Education, College of Biotechnology, Tianjin University of Science and Technology, Tianjin 300457, China; hyxiang97@mail.tust.edu.cn (Y.H.); 21816871@mail.tust.edu.cn (G.S.); fengxiaoqian@mail.tust.edu.cn (X.F.); niujialei@mail.tust.edu.cn (J.N.); wlu@mail.tust.edu.cn (L.W.); ycn21816865@mail.tust.edu.cn (C.Y.); luoxuegang@tust.edu.cn (X.L.); ma_wj@tust.edu.cn (W.M.); 2Qilu Institute of Technology, Jinan 250200, China

**Keywords:** obesity, *Kluyveromyces lactis*, *Lactobacillus plantarum*, gut microbiota, gut fungi

## Abstract

Obesity poses a significant threat to various health conditions such as heart diseases, diabetes, high blood pressure, and heart attack, with the gut microbiota playing a crucial role in maintaining the body’s energy balance. We identified a novel probiotic fungal strain, *Kluyveromyces lactis* JSA 18 *(K. lactis*), which was isolated from yak milk and was found to possess anti-obesity properties. Additionally, *Lactobacillus plantarum* CGMCC 8198 (LP8198) from our previous study was also included to evaluate its anti-obesity properties. The findings indicated that *K. lactis* caused a notable reduction in weight gain, liver and fat indexes, and hyperlipidemia in mice fed a high-fat diet (HFD). Administering *K. lactis* and LP8198 to mice on a high-fat diet resulted in a reduction of serum triglyceride levels. Furthermore, the supplements reduced ALT and AST activity, and inhibited the production of inflammatory cytokines such as TNF-α and IL-1β. In addition, lipid metabolism was enhanced by the downregulation of ACC1, PPAR-γ, SREBP-1, and Fasn. Moreover, this study found that *K. lactis* and LP8198 have little effect on gut bacteria. Additionally, *K. lactis* partially influenced intestinal fungi, while LP8198 had a minor influence on gut mycobiota. The main goal of this research was to show how effective *K. lactis* can be as a probiotic in combating obesity.

## 1. Introduction

Obesity refers to the excess storage of fat in the body, leading to the development of multiple health conditions like heart disease, diabetes, high blood pressure, stroke, and cancer [1]. Obesity has been categorized by the World Obesity Federation as a persistent, recurring disease that is among the most critical concerns in global public health, impacting over 650 million people. In developing countries, the problem of obesity is particularly severe. Diseases caused by obesity can have a detrimental impact on health and increase mortality [2,3,4]. Hence, it is imperative to have evidence-based techniques that can aid in achieving weight reduction.

The overconsumption of fatty foods is often linked to changes in the gut microbiome [5], which has a major impact on how the body absorbs energy [6]. It has been suggested that a decrease in gut microbial diversity and modifications in the population of certain microbes may be responsible for fat metabolism and obesity [7]. Moreover, the gut microbiota is involved in the progression of fat synthesis, absorption, and metabolism, which can lead to obesity and corresponding changes in the gut microbial community [8]. The presence of Bacteroides in normal individuals is advantageous, as they can break down dietary fiber for energy that the host is unable to digest or absorb. Nonetheless, when there is an accumulation of fat, this microbial community will alter, and the prevalence of Firmicutes in obese people will grow significantly [8,9,10]. Modifications to the community structure of intestinal microbes can inhibit the secretion of intestinal hormones, thereby reducing their impact on suppressing appetite, delaying gastric emptying, and increasing satiety [11].

With a profound knowledge of the gut microbiome, the functioning of probiotics and host microbiota is comprehensible today. Probiotics have the capability to generate antigens that are harmonious with the host, interact directly with the host’s immune cells, adjust host metabolism, etc. [12]. In experimental models of metabolic disease, *Lactobacillus plantarum* has been shown to have potential therapeutic effects on various conditions [13,14], such as improving hypertension [15], hypercholesterolemia [16], immunomodulation [17,18] and promoting fat loss and weight loss [19,20,21].

Yeast, the earliest microorganism utilized in human civilization, has been extensively investigated, and there are numerous studies both domestically and internationally that involve fermentation and strain improvement. The exploration of the probiotic activity of yeast commenced with the finding of Saccharomyces boulardi, and other yeasts with beneficial characteristics have been reported, such as *Saccharomyces cerevisiae* [22,23], *Kluyveromyces lactis* [24], *Debaryomyces hansenii*, *Geotrichum candidum*, *Yarrowia lipolytica* [25], and Pichia yeast. As research on yeast has progressed, it has been discovered that other yeasts possess probiotic properties, such as reducing cholesterol levels [26], inhibiting bacteria growth [27,28], exhibiting antitumor effects [29,30,31] and alleviating enteritis [32,33,34]. The results indicate that yeast could serve as an innovative probiotic, sparking a fresh avenue of investigation.

The current research investigated the weight-reducing properties of *Lactobacillus plantarum* CGMCC 8198 and *Kluyveromyces lactis* in combating obesity. Specifically, we examined the alterations in serum lipids, hormones, liver function, hepatic lipid accumulation, and epididymal fat expansion caused by these two probiotic strains. To elucidate the mechanisms underlying the anti-obesity effects, we thoroughly investigated the influences of these two strains on inflammatory response, gut microbiota composition, and gut metabolite profiles in HFD-induced mice. We also analyzed the relationships between these parameters. The findings from our study offer fresh perspectives on how *Lactobacillus plantarum* CGMCC 8198 and *Kluyveromyces lactis* contribute to reducing weight in obese mice induced by a high-fat diet. The results could aid in future studies investigating the use of probiotics as functional foods to prevent or treat obesity and its associated issues.

## 2. Materials and Methods

### 2.1. Strains, Cells, and Animals

#### 2.1.1. Strains

The strain JSA18 of *Kluyveromyces lactis*, sourced from yak milk, was confirmed as *Kluyveromyces lactis* using 16s RNA sequencing and is currently preserved at the China General Microbiological Culture Collection Center under the designation CGMCC No. 26714. *Lactobacillus plantarum* CGMCC 8198 (*Lactobacillus plantarum* TCCC 11824, LP8198) was probiotic-screened in our laboratory in the previous study [35,36]. *K. lactis* JSA18 was cultivated in YPD medium at 30 °C for a day, while *L. plantarum* CGMCC 8198 was cultured in MRS broth at 37 °C for the same duration. The strains were subsequently cultured three additional times to guarantee their viability and promote optimal growth. Afterward, the cells that were cultured were collected through centrifugation (10,000× *g*, 4 °C, 5 min) and rinsed three times with phosphate-buffered saline (PBS) for oral administration.

#### 2.1.2. Cells

NCM460 (laboratory, Tianjin, China), a well established human normal colonic epithelial cell line from healthy colon tissue samples, was employed in our experiments. The cells were cultured in Gibco’s Minimum Essential Medium, with the addition of 10% fetal bovine serum, at a temperature of 37 °C in a humid environment containing 5% CO_2_.

#### 2.1.3. Animals

Male C57BL/6J mice, aged four weeks and weighing between 15–17 g, were individually housed in a controlled environment at 24 ± 2 °C with a 12/12 h light/dark cycle. They were given unrestricted access to food and water for at least 7 days before the start of the experiments. The mice were obtained from ChangSheng Biology in Liaoning, China. The composition and formulation of the diets were carefully designed to ensure that the diets were nutritionally balanced and comparable in terms of taste and texture to minimize any discrepancies in palatability among the different diet groups. Afterward, the mice were assigned randomly to six groups, with 10 mice in each group (two cages for each group). Out of the 6 groups, one was given a standard diet (ND, 10 kcal% fat) (D12492, Research diet, New Brunswick, NJ, USA), another received a high-fat diet (HFD, 60 kcal% fat) (D12492, Research diet), and the remaining four groups were provided with a HFD that included 5 × 10^9^ CFU/mL of *K. lactis*, 5 × 10^9^ CFU/mL of LP8198, 5 × 10^9^ CFU/mL *K. lactis* + 5 × 10^9^ CFU/mL LP8198 (*v*/*v* = 1:1), or 10 mg/kg Orlistat. Each day, the mice were administered 200 μL of the respective treatment, with an average colony-forming unit (CFU) count of 1 × 10^9^. As a control placebo, the mice in the HFD group were administered 200 μL of PBS daily. The experiment was performed for 8 weeks. For the duration of the study, the body weights and food intake of the mice were observed weekly. Animal experiments were carried out following the Care Guidelines and approved by the Ethics Review Board for Animal Studies at Tianjin University of Science & Technology (approval number SWKL2022103009-30 October 2022).

### 2.2. Test for Tolerance to Acid and Bile

To guarantee its viability and optimal growth, *Kluyveromyces lactate* underwent three subcultures. Next, the sample was incubated at 30 °C for the entire night and then spun at 3000 rpm for a duration of 10 min. The collected bacteria were washed twice with PBS buffer, and the concentration was adjusted to 5 × 10^9^ CFU/mL. Next, 10 mL of synthetic stomach acid were added, and the mixture was incubated at 37 °C while being stirred at 150 rpm for 1.5 h. Following this, 17.5 mL of synthetic duodenal fluid was introduced, and the yeast was then placed in an incubator at 37 °C with a speed of 150 rpm for a duration of 3 h. Finally, the survival rate was assessed by growing on YPD plates. Artificial gastric juice formulation: 6.2 g/L NaCl, 2.2 g/L KCl, 0.22 g/L CaCl_2_, 1.2 g/L NaHCO_3_, 0.3% gastric enzyme, pH adjusted to 3.0. A synthetic mixture of duodenal fluid containing 6.4 g/L sodium bicarbonate, 0.239 g/L potassium chloride, 1.28 g/L sodium chloride, 0.1% pancreatic enzymes, and 10% bile salts, with a pH of 7.4.

### 2.3. Self-Cohesion, Hydrophobicity Experiments, and Adhesion of Kluyveromyces lactate to Cells

The *Kluyveromyces lactate* strain was passaged thrice to guarantee its viability and promote ideal development. Subsequently, it was left to incubate at 30 °C for the night and then subjected to centrifugation at 3000 rpm for 10 min. The collected *Kluyveromyces lactate* was then washed twice using PBS buffer. (i) The *Kluyveromyces lactate* was suspended in 3 mL of PBS, agitated for 30 s, and subsequently incubated at 37 °C. Subsequently, the optical density (OD) value was measured at 560 nm intervals of 0, 2, and 4 h. The rate of self-polymerization was determined using the formula self-polymerization (%) = A_t_/A_0_ × 100; (ii) The hydrophobicity of the *Kluyveromyces lactate* strain was assessed by adjusting the OD_560_ to 1.0 with PBS, then adding 1 mL of ethyl acetate and rotating for 120 s. The OD values before and after incubation were measured at 560 nm, and the hydrophobic rate of the strain was calculated using the formula: hydrophobic rate (%) = (A_0_ − A_t_)/A_t_ × 100.(A_0_: OD value before incubation; A_t_: OD value of different incubation times).

The adhesion assay was executed in accordance with the protocol that had been previously outlined [37]. NCM460 cells were placed in 96-well plates with a concentration of 1.0 × 10^8^ cells per well. After eliminating the medium, the plates were rinsed two times with PBS and subsequently placed in a 5% (*v*/*v*) CO_2_ environment with *Kluyveromyces lactate* for 1 h at 37 °C. Following incubation, any NCM460 cells not attached were eliminated by rinsing the cells with PBS two times before being examined using the MTT method. The survival rate was calculated based on the formula:Survival rate (%) = (OD_adhesion of *Kluyveromyces lactate*_ − OD_control_)/(OD_total *Kluyveromyces lactate*_ − OD_control_) × 100

### 2.4. Insulin Resistance (ITT) Analysis

Insulin resistance was measured in the seventh week of the animal experiment. Mice were fasting without water for 4 h, and fasting blood glucose was measured by tail tip blood sampling (0 min). The mice were then injected with insulin solution intraperitoneally at a dose of 0.75 U/kg (10 μL/g). The changes in blood glucose in mice were detected after injection for 15, 30, 60, and 120 min. The area under the curve was calculated according to the formula:AUG = (*G*_0_ + *G*_15_) + (*G*_15_ + *G*_30_) + (*G*_30_ + *G*_60_) + (*G*_60_ + *G*_120_)/2 × 30 (G: blood glucose values; 0, 30, 60, 90, 120: time (min)).

### 2.5. Blood, Liver, WAT, Cecum, and Feces Collection

Fecal samples were collected after the feeding experiment, following a 12 h period of fasting. Afterward, the mice were euthanized in order to obtain whole blood, the liver, epididymal adipose tissue (eAT), cecum, and feces. The blood was allowed to clot undisturbed at room temperature for 30 min, followed by centrifugation at 4 °C and 3000 rpm for 15 min to separate the clot from the serum. The serum was subsequently placed in a freezer at −80 °C, while the remaining tissues were quickly frozen in liquid nitrogen and also stored at −80 °C for future examination.

### 2.6. Serum Biochemical Parameter Analysis

Following the sacrifice of the mice, their blood serum was obtained through centrifugation, and the levels of total cholesterol (TC), triglycerides (TG), high density lipoprotein cholesterol (HDL-C), low density lipoprotein cholesterol (LDL-C), aspartate aminotransferase (AST), and alanine aminotransferase (ALT) were measured using commercially available kits according to the manufacturer’s guidelines (Jiancheng, Nanjing, China). Moreover, the levels of TNF-α and IL-1β in the serum were measured with ELISA kits (ZCIBIO Technology Co., Ltd., Shanghai, China).

### 2.7. Histological Analysis

Liver and epididymal white adipose tissue were treated with a 10% formalin solution, then encased in paraffin and sliced into 4 µm sections before being stained with H&E. Adipocyte size was examined using a light microscope from Olympus Optical Co. in Tokyo, Japan. The size was determined by measuring the area occupied by 20 adipocytes in a stained section. Additionally, Image J software (version 1.49) was employed to quantify the staining of the liver.

### 2.8. Primer Design for Quantitative PCR

The CDS sequences of target genes were found through the Genbank database, and the sequence was input into primer 5 software (version 6.0) to design primers. The primers were finally determined by setting the primer melting temperature (Tm) value range from 56 °C to 62 °C, setting the primer length range from 18 bp to 22 bp, and setting PCR product size range from 80 bp to 200 bp. Then, the appropriate primer pairs were chosen through exon/intron setting and primer specificity examination. The final primers are shown in Table 1.

### 2.9. Quantitative Real-Time PCR

Total RNA was extracted from liver cells by utilizing Trizol reagent (Invitrogen, Carlsbad, CA, USA), followed by reverse transcription of cDNA with random primers. The primers identified in Table 1 were utilized in quantitative real-time PCR to measure the relative gene expression levels, using Acta1 (β-Actin) as the reference gene. SYBR Green Master Mix from Invitrogen was utilized for quantitative real-time PCR analysis on a Bio-systems StepOneTM Real-Time PCR instrument from Applied Biosystems (Foster City, CA, USA), with cDNA serving as the templates. Three independent experiments were performed, each with three replicates.

### 2.10. Analysis of Fecal SCFAs

200 mg of fresh fecal samples were homogenized in 2 mL centrifuge tubes with ceramic beads using mechanical means and 1 mL of distilled water. After centrifuging the samples at 13,000 rpm for 10 min, the liquid portion was moved to a fresh tube. Afterward, a 10% solution of sulfuric acid and 1 mL of ethylacetate were combined in the tube, followed by vortexing and centrifugation at 13,000 rpm for 10 min. Afterward, the liquid portion was moved to containers for gas chromatography (GC) and kept in a freezer until examination with an Agilent 7890A GC (Agilent, Santa Clara, CA, USA).

The analysis of the samples was conducted with a HP-5 Polyethylene Glycol column (30 m × 320 μm × 0.25 μm) (Agilent). The GC was set with an injection temperature of 250 °C, an injection volume of 1 μL, and a split ratio of 1:1. The oven started at a temperature of 90 °C for 6 min, then increased to 200 °C over 10 min, and finally stayed at that level for another 6 min. The carrier gas was flowing at a rate of 2 mL/min. The flame ionization detector was adjusted to a temperature of 250 °C, with hydrogen and artificial air flowing at rates of 30 mL/min and 300 mL/min, respectively.

### 2.11. Microbial Community Analysis

(i) Bacteria. The samples were processed by Shanghai Majorbio Bio-Pharm Technology Co., Ltd. (Shanghai, China). Total DNA was extracted, amplified, and sequenced according to standard procedures. Briefly, microbial DNA was extracted from the cecal contents of the mice using the E.Z.N.A.^®^ Soil DNA Kit (Omega Bio-tek, Norcross, GA, USA), followed by measurement of the concentration with the Nanodrop device from Thermo Scientific, Waltham, MA, USA. Quality was assessed by agarose gel electrophoresis.16S rRNA gene sequence spanning the V3–V4 variable regions was amplified using the primers 338F (5′-ACTCCTAGGGAGGGCAGCAG-3′) and 806R (5′-GGACATCHVGGGTWTTCTAAT-3′). After extraction, the amplicons underwent purification with the AxyPrep DNA Gel Extraction Kit from Axygen Biosciences (Union City, CA, USA), followed by quantification using Quanti-FluorTM-ST from Promega (Madison, WI, USA). After combining equal amounts of the purified amplicons, the samples were subjected to paired-end sequencing with 2 × 300 reads using an Illumina MiSeq instrument (Illumina, San Diego, CA, USA).

(ii) Fungi. Shanghai Dasheng Biomedical Technology Co. (Dasheng, Shanghai, China) conducted internal transcribed spacer (ITS) sequencing to analyze fungal microbiomes. Total DNA was extracted from the cecum contents of the mice with the E.Z.N.A.^®^ soil DNA kit (Omega Bio-tek), and the DNA concentration and purity were assessed using a Nanodrop (Thermo Scientific) and agarose gel electrophoresis. The ITS2 region of the fungal ITS, which is highly variable, was amplified using a 20 μL PCR system and the primers ITS3F (5-GCATCGATGAAGAACGCAGC-3′) and ITS4R (5′TCCTCCGCTTATTGATATATGC-3′). PCR was performed with an initial denaturation at 95 °C for 3 min, followed by 28 cycles of 95 °C for 30 s, 55 °C for 30 s, and 72 °C for 30 s, and a final extension at 72 °C for 10 min. Subsequently, the PCR products were purified, and the amplicons were sequenced (200–500 bp) on the Illumina MiSeq platform (Illumina).

(iii) Raw sequencing reads underwent quality filtering using Trimmomatic and were subsequently merged by FLASH. Reads were truncated at sites with an average quality score < 20 over a 50 bp sliding window. Then, sequences with an overlap longer than 10 bp were merged if their overlap had no more than 2 bp mismatches. Sequences were sorted by sample-specific barcodes (requiring exact matches) and primers (allowing 2 nucleotide mismatches), while eliminating reads with ambiguous bases. Operational taxonomic units (OTUs) were clustered at a 97% for 16S and 98% for 18S similarity cutoff using UPARSE (version 7.1) with a novel ‘greedy’ algorithm that integrates chimera filtering and OTU clustering. The taxonomic classification of each 16S and 18S rRNA gene sequence was conducted using the RDP Classifier algorithm against the Silva (SSU123) 16S and 18S rRNA databases with a confidence threshold of 70%. Various indices (Shannon, Ace, and Chao) were calculated using distinct algorithms to estimate OTUs [38]. The above data analyses were all performed on the Majorbio I-Sanger Cloud platform (http://www.i-sanger.com, accessed on 12 March 2023).

### 2.12. Statistical Analysis

Experiment findings were displayed as mean ± standard deviation (SD) and assessed with SPSS 26.0 software via one-way ANOVA and two-way ANOVA. Significance was established based on a *p*-value below 0.05, denoted in the figure captions with symbols (* or #) to represent different levels of significance (*,^#^ = *p* < 0.05, **,^##^ = *p* < 0.01, ***,^###^ = *p* < 0.001).

## 3. Results

### 3.1. The Probiotic Strains K. lactis JSA18 and LP8198 Alleviated Body Weight Gain and Fat Accumulation in HFD-Fed Mice

*K. lactis* was retrieved from yak milk, which was then subjected to genome extraction and amplification via PCR. The sequencing of the extracted product was then compared to the Genebank database. The phylogenetic tree is depicted in Appendix A. Additionally, the probiotic properties of *K. lactis*, such as acid and bile salt tolerance and ad-herence properties, were preliminarily evaluated. As illustrated in Appendix A, the activity of *K. lactis* decreased by approximately 12% after 1.5 h in simulated gastric fluid and by approximately 33% after 3 h in simulated intestinal fluid, indicating that the strain has good acid and bile salt tolerance in the gastrointestinal tract. The cell adhesion ability of probiotics to NCM460 human normal colonic epithelial cells was assessed by the MTT method, and the self-polymerization rate and hydrophobic rate of the strain were also examined, as indicated in Appendix A. This was performed to determine the adhesion of probiotics to intestinal epithelial cells, which is facilitated by a strong hydrophobic interaction force. The strain displayed a high self-polymerization rate of 87.7% after 4 h, a hydrophobic rate of 52.3% after 1 h, and an adhesion rate of 44.5% to NCM460 cells, indicating good adhesion capabilities. In conclusion, due to its acid and bile salt resistance and strong adhesion to cells, *K. lactis* is suitable for use as a probiotic.

To assess the anti-obesity effects of *K. lactis* and *K. lactis* supplemented with LP8198, both normal and high-fat diets were administered to the subjects for a duration of 8 weeks, with a focus on the variations in body weights between the ND and HFD groups (Figure 1A). To further assess the effects of two probiotic strains, *K. lactis* and LP8198, they were added to the HFD groups. The daily food intake was observed, and no significant difference was found among the groups (Figure 1B). The food efficiency ratio (FER) provides an effective parameter for predicting the weight loss effect. The FER of the HFD group was approximately 2 times that of the ND group, while the supplementation of *K. lactis* significantly reduced the FER (Figure 1C). The *K. lactis* group showed a 26.8% decrease in body weight gain compared to the HFD group after 8 weeks, with a weight gain of 7.42 ± 0.94 g for *K. lactis* and 10.13 ± 1.26 g for HFD. The orlistat group had a 20.4% lower weight gain than the HFD group (Figure 1D). Furthermore, we evaluated the weight of the liver and epididymal adipose tissue (eAT) in mice from all groups. Findings indicated that the groups treated with *K. lactis*, LP8198, *K. lactis* + LP8198, and orlistat all showed a significant reduction in liver and eAT weight in comparison to the HFD group (Figure 1E,F). Obesity is known to be a factor in the emergence of diabetes. In this study, a high-fat diet results in abnormal blood glucose levels in mice, while *K. lactis*, LP8198, and the combination of *K. lactis* and LP8198 intervention were found to be successful in decreasing the blood glucose levels (Figure 1G), thus preventing the occurrence of diabetes. Moreover, the Insulin Tolerance Test (ITT) showed that the insulin resistance of the HFD group was significantly higher than that of the ND group, and the insulin resistance of *K. lactis*, LP8198, and the combination of *K. lactis* and LP8198 groups was alleviated (Appendix A). The findings indicate that *K. lactis* and LP8198 have the potential to greatly decrease weight gain, liver size, fat levels, and fasting blood glucose levels in HFD mice.

### 3.2. The Probiotic Strains Showed an Alleviation of Physiological Changes Caused by a HFD

In order to assess how probiotic strains affect lipid metabolic disorders resulting from a high-fat diet, a study was carried out to analyze four blood lipid parameters (total cholesterol (TC), serum triglyceride (TG), high-density lipoprotein cholesterol (HDL-C), and low-density lipoprotein cholesterol (LDL-C)) in plasma (Figure 1H–K). The findings showed that the groups receiving *K. lactis*, LP8198, and *K. lactis* + LP8198 interventions, as well as orlistat, were able to significantly prevent the elevation of TG induced by HFD (Figure 1I). The concentrations of TC were increased in the HFD group, while they were not affected by *K. lactis*, LP8198, or *K. lactis* + LP8198 interventions (Figure 1H). Nevertheless, there was no notable variation in the HDL-C and LDL-C concentrations between the ND, HFD, and probiotic strain groups (Figure 1J,K).

The liver is the primary metabolic organ of the body and is responsible for the production of lipids. Obesity may result in nonalcoholic fatty liver disease and other associated conditions, marked by enlarged lipid droplets and reduced intercellular space, causing compression of neighboring cells [39]. The histopathological morphologies of the liver and eAT were analyzed with H&E staining to assess the preventive impact of probiotic strains and orlistat on liver and eAT injury caused by a HFD. The ND group displayed a clear hepatic cellular structure (Figure 1L), whereas the HFD group showed accumulation of lipid droplets and infiltration of inflammatory cells (Figure 1L). Conversely, the lipid droplets and lesion area were notably reduced in the *K. lactis*, LP8198 + *K. lactis*, and orlistat groups compared to the HFD group (Figure 1L and Appendix A). The LP8198 + *K. lactis* group demonstrated the most effective outcome in mitigating liver damage. H&E staining of eAT slides revealed that the adipocytes in mice fed a HFD were notably larger than those in the ND group. Yet, the addition of *K. lactis* and LP8198 plus *K. lactis* effectively inhibited the increase in adipocyte size induced by the HFD (Figure 1M and Appendix A).

The results suggest that HFD consumption can induce obesity and hyperlipidemia, as shown by multiple factors such as liver and eAT levels, yet *K. lactis* and LP8198 plus *K. lactis* can alleviate this progression.

### 3.3. The Probiotic Strains Showed a Positive Effect on Liver Damage and Inflammation in the Serum Caused by a High-Fat Diet

In order to enhance the comprehension of how probiotic strains impact liver damage and systemic inflammation, cytokine expression in the serum was assessed. High levels of ALT and AST in the blood indicated the early stages of liver damage. As shown in Figure 2A,B, ALT and AST levels were markedly elevated in the HFD group compared to the ND group, suggesting liver injury in the HFD group. Nevertheless, probiotic strains and orlistat treatment were found to be effective in reducing liver injury (Figure 2A,B). It is well known that obesity can activate the immune response in adipose tissue, and inflammation is also seen in other organs. In order to examine this, the levels of inflammatory proteins like TNF-α and IL-1β in the blood were measured in different sets of mice. The research showed that the HFD groups had notably elevated levels of TNF-α and IL-1β in their serum, in contrast to the ND groups. Treatment with probiotic strains and orlistat led to a significant decrease in TNF-α and IL-1β levels (Figure 2C,D).

The results indicate that HFD caused liver damage and inflammation, which was alleviated by the administration of *K. lactis*, LP8198, and a combination of both.

### 3.4. The Probiotic Strains Have an Influence on the Transcription of Genes Related to Lipid Metabolism

The impact of probiotic strains LP8198 and *K. lactis* on obesity was assessed by examining the mRNA expression of genes related to lipid metabolism in the liver. The study findings showed that there was a notable increase in the expression of genes associated with fat production, including PPAR-γ, SREBP-2, and fatty acid synthase (Fasn), in the high-fat diet group compared to the normal diet group. In contrast, the probiotic strains and orlistat groups showed a notable decrease in the mRNA expression levels of these genes (Figure 3D–F). In the HFD group, the mRNA levels of genes related to the oxidation of fatty acids, like PPARα and CPT1, were lower than those in the ND group. Nevertheless, the gene expression levels of PPARα and CPT1 did not show any significant difference after probiotic strains and orlistat treatment (Figure 3A,B). In addition, our research showed that the mRNA levels of the acetylCoA carboxylase1 Acaca (ACC1) gene, which is related to the fatty acid biosynthetic process, increased significantly in the fat tissue of mice on a high-fat diet. However, the level of ACC1 decreased significantly compared to the HFD group after the mice were given probiotics and orlistat. (Figure 3C).

The findings suggest that LP8198 and K. lactis have the ability to influence the activity of genes associated with fat production, showcasing their potential to combat obesity.

### 3.5. The Effects of Probiotic Strains on Short-Chain Fatty Acids (SCFAs) Metabolic Profiling

SCFAs are the primary microbial metabolites and are essential for regulating the host’s metabolism. To investigate the effects of *K. lactis* and LP8198 on SCFA metabolic profiling, the cecum samples were analyzed quantitatively for the six short-chain fatty acids: acetic, propionic, butyric, isobutyric, valeric, and isovaleric acids. As demonstrated in Figure 4, the levels of acetic, propionate, isobutyric, valeric, and isovaleric were notably reduced in the HFD group in comparison to the ND group, with butyric levels remaining relatively unchanged. Treatment of the HFD group with *K. lactis*, LP8198, and LP8198 + *K. lactis* resulted in a significant rise in SCFAs, acetic acid, isobutyrate, and valeric acid concentrations in fecal samples (Figure 4). It was observed that the levels of five short-chain fatty acids (excluding butyric acid) in the orlistat group had increased, while the LP8198 + *K. lactis* group experienced an increase in propionate acid, valeric acid, isobutyric acid, and isovaleric acid. The results suggest that certain probiotic bacteria can increase the production of short-chain fatty acids, potentially benefiting obese mice on a high-fat diet.

### 3.6. The Probiotic Strains Have an Effect on the Gut Microbial Composition of Mice That Are Fed with a High-Fat Diet

We analyzed the impact of probiotic strains LP8198 and *K. lactis* on the gut microbiota by sequencing the V3–V4 region of 16S rRNA genes from cecal content samples and utilizing alpha diversity metrics like the Ace index, Chao1 index, and Shannon index to assess the microbiota’s abundance and diversity. Based on Figure 4A–C, the Ace and Chao1 metrics indicated a slight decrease, but not significant, in microbial diversity due to HFD, whereas the Shannon metric suggests an increase in intestinal microbiome richness compared to the ND group. However, the richness of the intestinal microbiome showed a slight, though not significant, increase in the HFD + *K. lactis* and HFD + LP8198 groups when compared to the HFD group (Figure 5A,B). The little impact of HFD, *K. lactis*, and LP8198 on the richness and diversity of the gut microbiota in mice was shown by the Shannon index (Figure 5C).

Principal Coordinate Analysis (PCoA) using the Bray–Curtis dissimilarity metric was utilized to assess the overall variation in gut microbiota structure across various groups. In general, there was a clear distinction between the HFD group and ND group, with the HFD + *K. lactis* group, HFD + LP8198 group, and HFD + *K. lactis* + LP8198 treatment groups showing some deviation from the HFD group (Figure 5D). The findings indicated that the gut microbiome of mice fed a high-fat diet was changed, with *K. lactis* and LP8198 impacting the gut microbiome of the high-fat diet group (Figure 5E,F). PCoA analysis also revealed an overlap between the HFD alone group and the HFD + *K. lactis*/HFD + *K. lactis* + LP8198 groups (Figure 5E–H), while there was a dispersed distribution between the HFD + LP8198 group and HFD group (Figure 5E–H). This indicates that the impact of LP8198 on the intestinal flora of mice fed a high-fat diet was more significant.

### 3.7. The Effect of HFD, K. lactis, and LP8198 Exposure on the Composition of the Intestinal Microbiome

A stacked histogram was used to assess how each group impacted the specific makeup of gut bacteria at the portal level. Exposure to a high-fat diet was found to alter the distribution of bacterial communities at the phylum level (Figure 6A), although these changes were not statistically significant (Figure 6B). Firmicutes, Bacteroidetes, Actinobacteria, and Verrucomicrobia were the dominant phyla in mice at the phylum level, with their proportions staying consistent in the HFD, HFD + *K. lactis*, HFD + LP8198 + *K. lactis*, and HFD + orlistat groups. The only exception was Verrucomicrobia, which showed a notable rise in the orlistat group. In comparison to the ND group, the relative abundance of Desulfobacterota and Proteobacteria was little changed in the HFD group, while the LP8198 + *K. lactis* and orlistat groups showed a notable decrease in Proteobacteria abundance (Figure 6B). The family Patescibacteria remained unchanged in the HFD, HFD + *K. lactis*, HFD + LP8198, and HFD + LP8198 + *K. lactis* groups. Additionally, the families Campilobacterota and Deferribacterota did not show any significant differences across all groups (Figure 6B). Results suggest that HFD, *K. lactis,* and LP8198 may have a minor impact on the composition of the intestinal microbiome.

### 3.8. The Probiotic Bacteria Change the Gut Fungal Composition in Mice Consuming a High-Fat Diet

Moreover, *K. lactis* is a fungus, so we investigated the effects of HFD, LP8198, and *K. lactis* on intestinal flora by high-throughput sequencing analysis of fungal ITS rRNA genes in the cecum contents of six groups. In this study, α-diversity indices such as the Ace index, the Chao1 index, and the Shannon index were used to evaluate the abundance and diversity of fungal species. Figure 7A–C demonstrates that the microbial community abundance, as revealed by Ace, Chao, and fungal indices, was slightly higher in the HFD group. The Ace and Chao indices, however, indicated a decrease in intestinal microbial diversity following *K. lactis* and Orlistat treatment, with LP8198 having a negligible impact on community diversity (Figure 7A–C).

Principal Coordinate Analysis (PCoA) using Bray–Curtis distance was used to examine the overall differences in structure of the gut microbiota among various groups. The findings showed clear distinctions between the HFD, orlistat, and ND groups, with the three treatment groups deviating to some extent from the HFD group (Figure 7D). PCoA was employed to assess the impact of different treatment doses on the gut microbiota of mice fed a high-fat diet (HFD) (Figure 7E–H). Compared to the HFD group, the *K. lactis* group was the most disparate, indicating that its intestinal flora was the most dissimilar from the HFD group, with the orlista group following, and there were different contact distances between the LP8198 + *K. lactis* groups.

The results showed that *K. lactis* had a more prominent influence on the gut fungal composition of HFD-fed mice, whereas LP8198 had a negligible effect.

### 3.9. The Influence of Exposure to HFD, K. lactis, and LP8198 on the Composition of Intestinal Fungi

The findings from Figure 8A indicate that the presence of fungal genera could be influenced by a diet high in fat. In particular, Wallemia, Kazachstania, Penicillium, and Aspergillus were the most dominant genera in mice, and their relative abundance was notably different between the ND and HFD groups, without any significant changes in the HFD + LP8198 and HFD + *K. lactis* groups (Figure 8B). Penicillium and Aspergillus showed increased expression in the HFD groups and HFD treatment groups when compared to the ND group, while Wallemia exhibited a slight rise in the HFD group compared to the ND group. Additionally, LP8198 + *K. lactis,* and orlistat were observed to reduce the content of Wallemia in the HFD group. Finally, there was a notable reduction in Kazachstania levels in the HFD group and HFD + probiotic strains when compared to the ND group (Figure 8B). Furthermore, there was a slight decrease in Rhizobium levels in the HFD group compared to the ND group. However, Rhizobium levels notably rose in the *K. lactis* and Orlistat groups when compared to the HFD group (Figure 8B). The Naganishia level was increased by HFD, while probiotic strains and orlistat could restore its level to that of the ND group (Figure 8B). This implies that LP8198 and *K. lactis* had a minimal effect on the gut fungal composition of HFD-fed mice.

## 4. Discussion

Obesity is characterized by the buildup of too much fat in tissues, resulting in a higher mass of adipose tissue, and is a significant factor in metabolic disorders like high cholesterol and fatty liver disease [1,40]. Despite the use of pharmacological approaches to treat obesity, their effectiveness is limited due to the possibility of adverse side effects. Therefore, probiotic administration has been suggested as an alternative option [41,42]. The objective of this research was to investigate how LP8198, *K. lactis*, and their combination affect weight reduction in obese mice induced by a high-fat diet. The findings indicated that the probiotics had a notable impact on reducing obesity caused by a high-fat diet, as evidenced by the decrease in body weight and organ and epididymal fat levels (Figure 1A,D–F). Moreover, there was no significant difference in food intake among the groups (Figure 1B). We speculated that probiotic treatment might induce increased fecal discharge, leading to more energy consumption and subsequent weight loss. Future studies should assess digestibility and energy absorption by measuring energy loss in feces to confirm this hypothesis. Additionally, examination with H&E staining showed that the probiotics led to a reduction in both the quantification of liver pathologies and injury and the size of adipocytes when compared to the high-fat diet group (Figure 1L,M and Appendix A).

The expansion of adipocytes is thought to play a significant role in obesity, leading to the buildup of fat and a subsequent rise in levels of inflammation [43,44]. Macrophages, which are responsible for inflammation, can express surface markers and secrete proinflammatory cytokines, thus transforming local adipose tissue inflammation into a systemic one [45,46]. LP8198 and *K. lactis* are believed to be effective in preventing obesity by suppressing fat storage in adipose tissue and reducing chronic inflammation in the same tissue (Figure 1 and Figure 2). Additionally, increased amounts of overall cholesterol and blood sugar are frequently linked to obesity and dyslipidemia, conditions that have been seen in experiments involving diet-induced obesity [47]. In our research, it was demonstrated that LP8198 and *K. lactis* were able to lower serum TG levels in obese mice induced by a high-fat diet, as shown in Figure 1I. Although serum lipid levels reflected hepatic lipid metabolism to some extent, more direct measurements of hepatic lipids (intrahepatic triglyceride and intrahepatic cholesterol) should be performed in future studies. This would provide a better understanding of the role of hepatic lipids regulation in the influence of probiotics on obesity. Additionally, fatty acids from adipose tissue were found to be released into the liver due to obesity, which could lead to the development of a fatty liver and dyslipidemia in the bloodstream [48,49,50]. These findings suggest that LP8198, *K. lactis*, and their combination can improve dyslipidemia.

The liver plays a crucial function in the body by synthesizing, metabolizing, storing, and distributing carbohydrates, proteins, and lipids [51]. The production of lipids is mainly controlled by two key enzymes that limit the rate of synthesis: Fasn and ACC1. Fasn is governed by PPAR-γ’s positive feedback loop, while ACC1 is regulated by PPAR-γ’s negative feedback regulation. Activation of PPAR-γ triggers a heightened expression of proteins related to fatty acid transport and an increase in cellular triglyceride synthesis [52]. Additionally, when the body of AMPK is activated, PPAR expression is increased, which suppresses the expression of ACC1 and concurrently boosts the expression of CPT1. As a result, there is a rise in the breakdown of lipids through fatty acid beta oxidation in hepatic cells [52,53]. The study findings suggest that administering LP8198 and *K. lactis* can lower the expression of genes related to fat formation in obese mice on a high-fat diet, including PPAR-γ, SREBP-2, Fasn, and ACC1, while increasing PPAR and CPT1 levels (Figure 3).

Our findings suggest that LP8198 and *K. lactis* supplementation were associated with improvements in lipid metabolism and weight loss. However, further studies are required to elucidate the underlying mechanisms and establish a direct causal relationship between these probiotics and the observed metabolic changes. The circulation of free fatty acids throughout the body can lead to an increase in the production of liver fats and the initiation of reactive oxygen species formation. When the reactive oxygen species content surpasses the antioxidant system’s capacity to eliminate them, lipid peroxidation will occur in the liver, resulting in disruption of mitochondria and lysosomes’ structure, abnormal cell energy metabolism, and even cell disintegration [54]. This study revealed that the obese model group had severe liver damage, while the probiotic intervention group showed varying degrees of improvement (Figure 1).

Certain types of gut bacteria can break down carbohydrates like low-digestible polysaccharides and dietary fiber to create SCFAs like acetic, propionic, and butyric acid [55,56]. SCFAs are both a source of energy and a signaling molecule that can have a positive impact on the host’s health and immune system [57]. Research has shown that butyric acid could potentially help prevent various conditions, including colorectal cancer, inflammatory bowel disease, diabetes, and obesity [58,59]. Rodent models have demonstrated that the consumption of butyric acid and acetic acid may inhibit weight gain [60]. Additionally, studies have shown that consuming butyric acid and propionic acid can decrease food consumption and weight gain from a high-fat diet while also preventing abnormal glucose tolerance [60]. Furthermore, the positive effects of butyric acid on homeostasis and energy metabolism in the gut have been documented [11,61]. In this study, the different amounts of short-chain fatty acids in the four intervention groups may be one of the reasons for the differences in the regulation of obesity and glycolipid metabolism (Figure 4).

Studies over the last ten years have shown a connection between obesity, changes in diet, and shifts in the makeup of gut bacteria [62], with gut fungi playing a vital part in human well-being and different illnesses linked to the formation of gut bacteria. The three major bacterial phyla in the human intestinal tract, Firmicutes, Bacteroidetes, and Actinobacteria, account for more than 90% of all bacteria. Along with bacteria and viruses, a significant number of fungi inhabits the human gut, making up 0.1% of all gut microorganisms. It is thought that these fungi play a role in balancing the gut and the development of conditions [63,64] like autoimmune [65,66,67], metabolic [68,69,70], neurological [71,72], and cancer [73,74] diseases. We observed that *K. lactis* had less effect on microbial diversity than LP8198, possibly because *K. lactis*, as a fungus, played a role in the gut microbiota through different mechanisms than bacteria and may have a less direct impact on bacterial populations (Figure 5E,F). Nevertheless, the LP8198 and *K. lactis* interventions in this study did not show any significant impact on the bacteria in the intestinal microbiota (Figure 5 and Figure 6). Previous research has revealed several potential mechanisms through which probiotics might influence weight and lipid metabolism by directly modulating host metabolic pathways involved in lipid metabolism and energy balance, regulating immune function, enhancing gut barrier function, modulating neuroendocrine pathways, and producing bioactive metabolites [75,76]. Additionally, probiotics could directly impact liver function, improving hepatic lipid clearance and metabolism [76]. We hypothesize that LP8198 and *K. lactis* may exert their effects on body weight and lipid metabolism through one or more of these mechanisms, rather than by significantly altering the gut microbiota composition in mice.

The gut mycobiota, although present in low abundance, is gaining recognition for its role in maintaining health and its links to various diseases such as primary sclerosing cholangitis, colorectal cancer, and allergic inflammation [77,78]. Clearly, fungi play a crucial role in the gut microbiome. In this study, we further investigated the effects of fungi by observing the changes in the gut of mice with HFD, LP8198, and *K. lactis* interventions. Short-chain fatty acids were measured in feces to assess microbial metabolic activity throughout the colon and rectum, while fungal microbiota was analyzed in cecal contents to directly investigate the distribution and activity of fungi in the digestive system, as the cecum provides a favorable environment for fungal growth. Results indicated that the abundance of the fungi community was slightly increased with the HFD diet, whereas *K. lactis* and Orlistat treatments reduced the intestinal fungi diversity (Figure 7). Despite the intervention of *Lactobacillus plantarum* LP8198, the diversity of fungi remained unaffected (Figure 7). Moreover, the relative abundance of fungi was also monitored and evaluated. It was observed that the abundance of Rhizopus was significantly reduced, while the abundance of Fusarium and Naganishia increased after HFD feeding. Following treatment with LP8198 and *K. lactis*, the abundance of Rhizopus was enhanced, while the relative abundances of Fusarium and Naganishia were reduced. These findings suggest that the HF diet has an effect on the abundance of some fungi, and LP8198 and *K. lactis* interventions partially recover the abundance of these fungi (Figure 8). Conversely, most fungi were not influenced by the HFD diet or probiotic treatment (Figure 8).

Although our study demonstrated the individual effects of *K. lactis* and LP8198 on body weight and lipid metabolism, we also noted a lack of synergistic effects when they were used in combination. This observation highlights the complexity of microbial interactions and the various factors that may influence their outcomes. Firstly, microbial interactions can be influenced by factors such as competition, cooperation, and their impact on the host gut environment. Secondly, differences in dosage and administration methods in our experimental design may have contributed to the underrepresentation of synergistic effects between *K. lactis* and LP8198. Furthermore, individual host responses to microbial combinations may vary due to factors such as genotype, dietary habits, and metabolic status. While significant synergistic effects were not observed in our study, this finding provides impetus for further exploration of microbial combination therapies to better understand their mechanisms and maximize therapeutic efficacy.

## 5. Conclusions

The current research investigated the impact of LP8198 and *K. lactis* on obesity in mice fed a high-fat diet. Both varieties were successful in significantly reducing body weight and fat mass while also enhancing lipid storage in the liver and decreasing the size of epididymal fat, liver damage, and inflammation. Moreover, these probiotics were found to regulate AMPK pathway-related genes, resulting in the alteration of PPAR-γ, SREBP-2, Fasn, and ACC1 expression levels. Furthermore, the treatment of these two probiotics can also affect the changes in SCFAs through the changes in intestinal flora in the HFD group. Taken together, these findings imply that, aside from LP8198, *K. lactis* can be employed to both cure and avert obesity.

## Figures and Tables

**Figure 1 foods-13-01124-f001:**
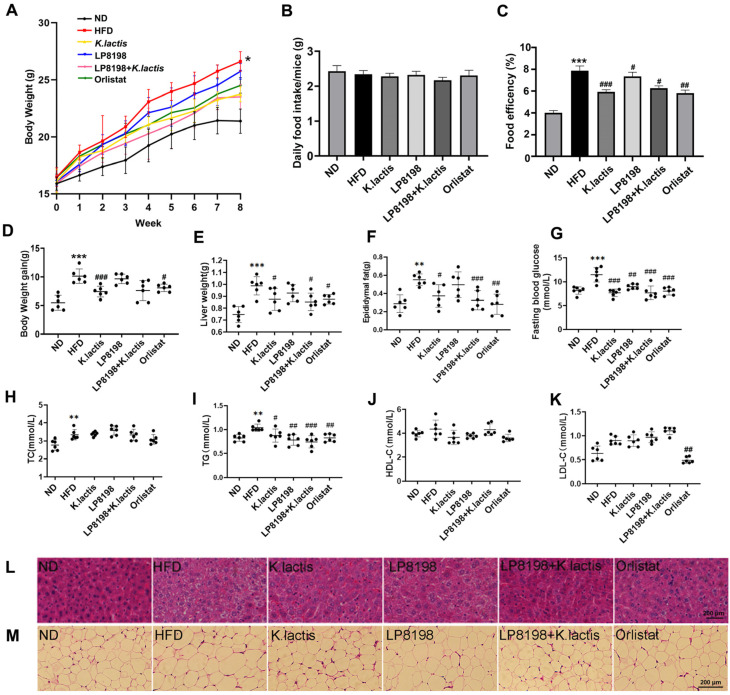
Impact of probiotic strains on weight gain and parameters related to obesity in mice fed a high-fat diet. (**A**) Alterations in the body weight of mice were observed over the course of an 8-week period during which probiotic strains were administered. Statistical significance was evaluated using a two-way ANOVA followed by a Dunnett’s post hoc test. * *p* < 0.05 compared to the ND group. (**B**) The daily food intake of mice treated with HFD and probiotic strain treatments for 8 weeks. (**C**) The food efficiency of mice treated with HFD and probiotic strain treatments for 8 weeks. (**D**) Following an 8-week probiotic strains treatment, the mice’s weight gain was monitored. (**E**) After 8 weeks of probiotic strain intervention, the liver weight of mice was observed. (**F**) Following an 8-week period of administering probiotic strains, the weight of epididymal adipose tissue (eAT) in mice was measured. (**G**) Blood glucose levels after fasting; (**H**) total cholesterol levels in the blood (TC). (**I**) Serum triglyceride (TG). (**J**) Serum high-density lipoprotein cholesterol (HDL-C). (**K**) Serum levels of low-density lipoprotein cholesterol (LDL-C). Statistical significance was evaluated using one-way ANOVA followed by a Dunnett’s post hoc test for (**B**–**K**). * *p* < 0.05, ** *p* < 0.01, and *** *p* < 0.005 compared to the ND group; ^#^
*p* <0.05, ^##^
*p* <0.01, and ^###^
*p* < 0.005 compared to the HFD group. (**L**,**M**) Representative H&E stained images of eAT and liver from subjects. ND: normal diet; HFD: high-fat diet; *K. lactis*: HFD + *K. lactis*, LP8198: HFD + LP8198; LP8198 + *K. lactis*: HFD + LP8198+ *K. lactis*; Orlistat: HFD+ Orlistat. Mean ± SD data from six mice in each group are presented.

**Figure 2 foods-13-01124-f002:**
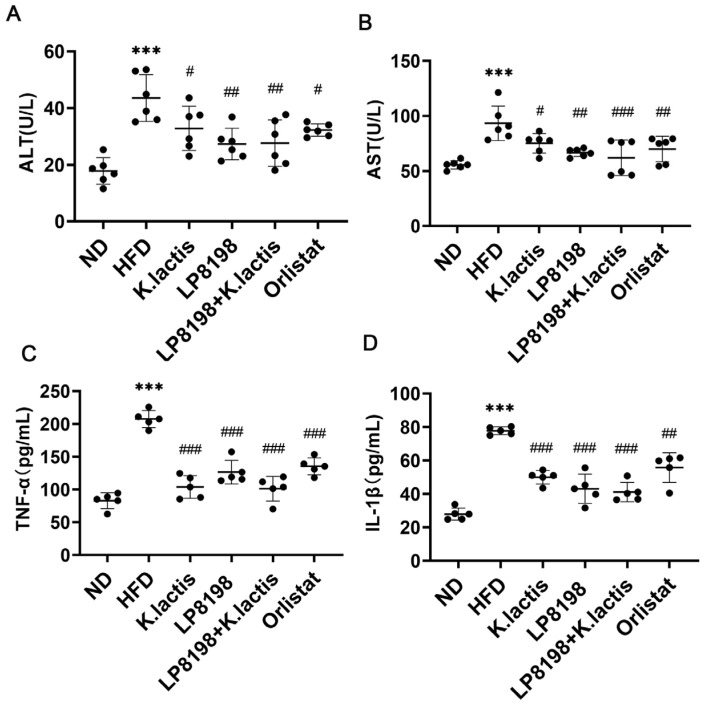
Impact of probiotic varieties on serum markers of liver damage and inflammation in mice fed a high-fat diet. (**A**,**B**) Serum levels of alanine aminotransferase (ALT) (**A**) and aspartate aminotransferase (AST) (**B**) were measured to assess liver health. (**C**,**D**) Serum levels of TNF-α (**C**) and IL-1β (**D**) inflammation markers were measured, with data presented as the average ± standard deviation (*n* = 5). The data is presented as the average plus or minus the standard deviation, with a sample size of 6. Statistical significance was evaluated using one-way ANOVA followed by a Dunnett’s post hoc test. *** *p* < 0.005 compared to the ND group; ^#^
*p* < 0.05, ^##^
*p* < 0.01, and ^###^
*p* < 0.005 compared to the HFD group.

**Figure 3 foods-13-01124-f003:**
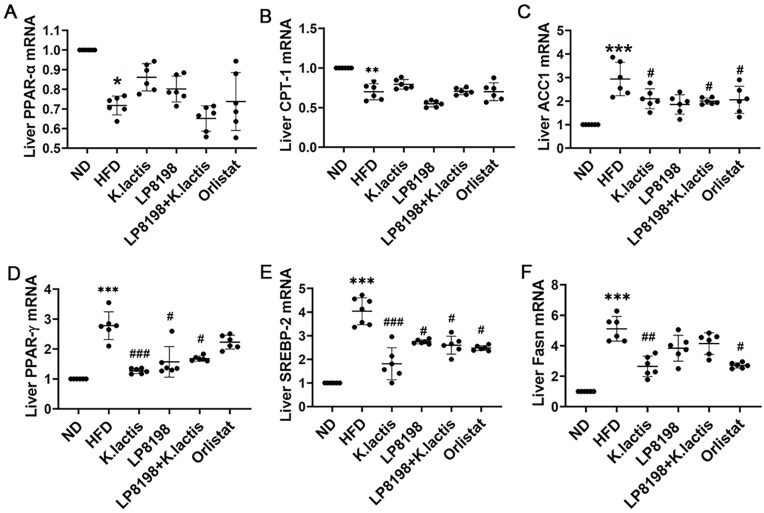
The impact of introducing probiotic strains on the expression of genes associated with lipid processing in the livers of mice fed a high-fat diet. (**A**–**F**) The mRNA levels of genes associated with the oxidation of fatty acids were measured. Mean ± SD values are presented (*n* = 6). Statistical significance was evaluated using a one-way ANOVA followed by a Dunnett’s post hoc test. * *p* < 0.05, ** *p* < 0.01, and *** *p* < 0.005 compared to the ND group; ^#^
*p* < 0.05, ^##^
*p* < 0.01, and ^###^ *p* < 0.005 compared to the HFD group.

**Figure 4 foods-13-01124-f004:**
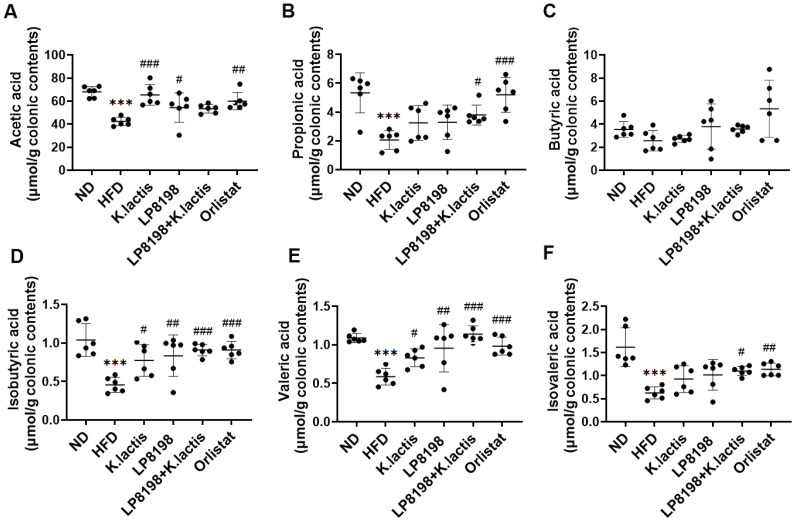
The influence of probiotic varieties on the metabolic profile of SCFAs in the fecal matter of mice consuming a high-fat diet. (**A**) acetic acid, (**B**) propionic acid, (**C**) butyric acid, (**D**) isobutyric acid, (**E**) valeric acid, and (**F**) isovaleric acid are listed as options. Mean ± SD values are presented (*n* = 6). Statistical significance was evaluated using a one-way ANOVA followed by a Dunnett’s post hoc test. *** *p* < 0.005 compared to the ND group; ^#^
*p* < 0.05, ^##^
*p* < 0.01, and ^###^ *p* < 0.005 compared to the HFD group.

**Figure 5 foods-13-01124-f005:**
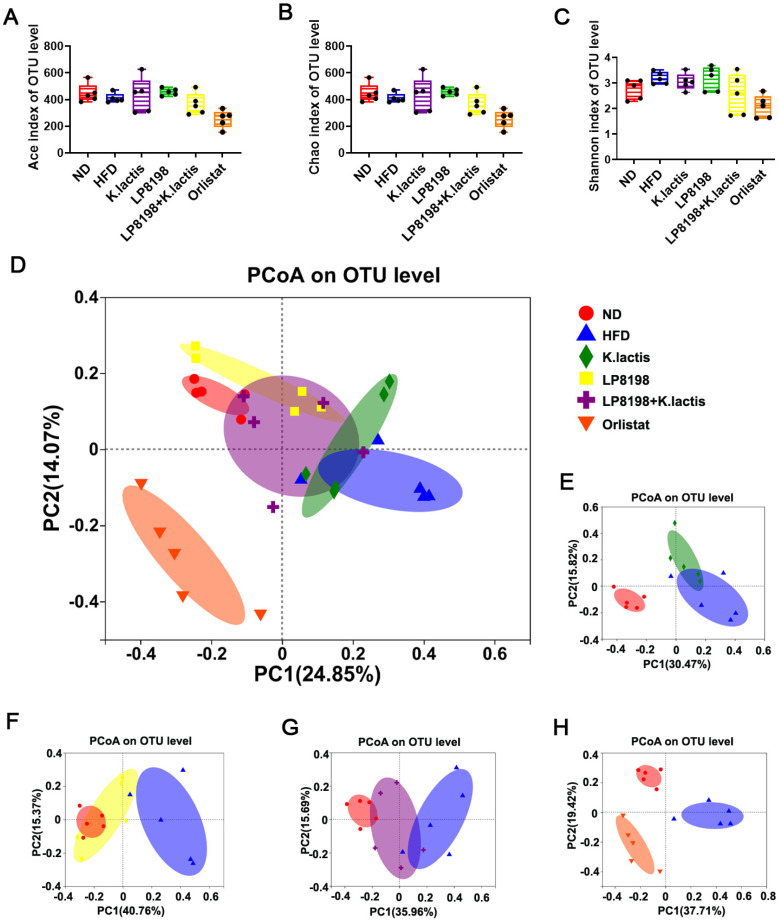
Changes in the gut microbiota of probiotic strains were observed in obese mice induced by a high-fat diet. (**A**) Alpha diversity was evaluated by Ace. (**B**,**C**) Alpha diversity was assessed using Chao1 and Shannon indices. (**D**,**E**) Principal coordinate analysis (PCoA) was conducted based on the Bray–Curtis distance at the OTU level for all mice, as well as for mice in the ND, HFD, and HFD + *K. latis* groups. (**F**–**H**) PCoA analysis using the BrayCurtis distance was conducted on mice at the OTU level in different groups, including ND, HFD, HFD + LP8198, and HFD+ LP8198 + *K.latis*, as well as HFD + orlistat. The data is presented as the average plus or minus the standard deviation, with a sample size of 5.

**Figure 6 foods-13-01124-f006:**
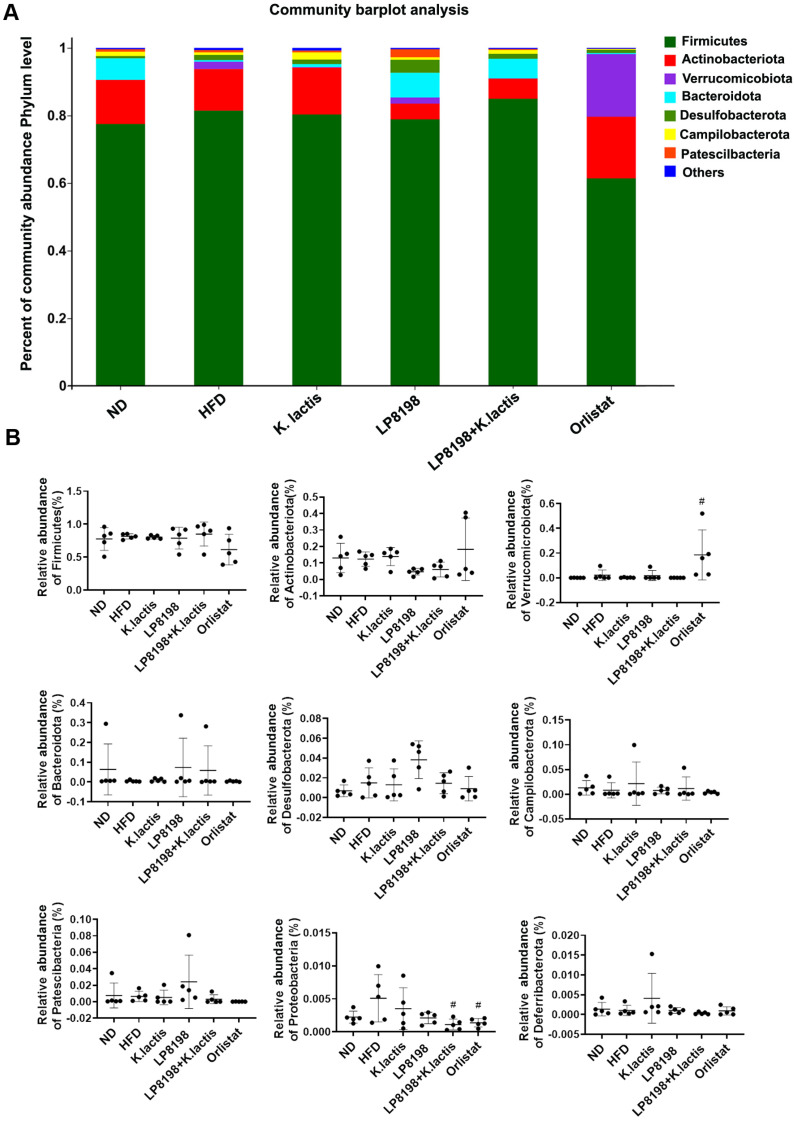
Examining the impact of a high-fat diet on the phylum level composition of gut microbes in the colon. (**A**) Bacterial taxonomic profiling at the phylum level. (**B**) The relative abundance of the gut microbiota at the phylum level. Statistical significance was evaluated using a one-way ANOVA followed by a Dunnett’s post hoc test. ^#^
*p* < 0.05 compared to the HFD group.

**Figure 7 foods-13-01124-f007:**
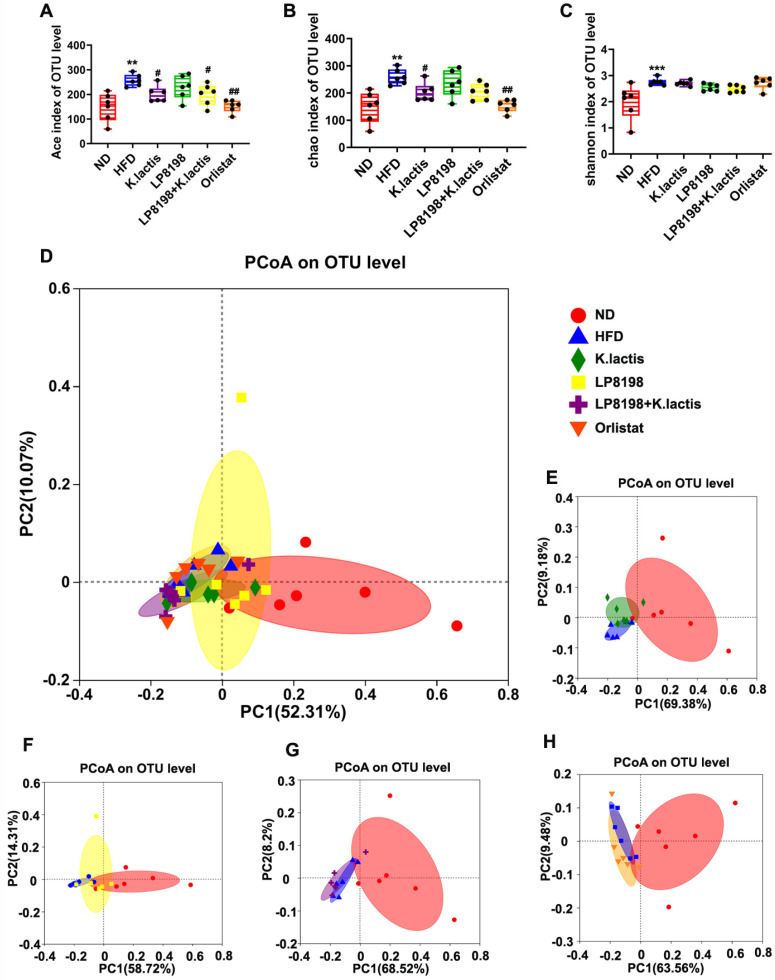
Changes were noted in the fungal population of the cecum in obese mice induced by a high-fat diet when probiotic strains were introduced. Ace was used to measure Alpha diversity. Alpha diversity was measured by Chao1, and (**C**) the Shannon diversity index calculated Alpha diversity. Statistical significance was evaluated using a one-way ANOVA followed by a Dunnett’s post hoc test for (**A**–**C**). ** *p* < 0.01, and *** *p* < 0.005 compared to the ND group; ^#^
*p* < 0.05 and ^##^
*p* < 0.01 compared to the HFD group. (**D**) Principal Coordinates Analysis (PCoA) utilized the BrayCurtis dissimilarity metric for all mice at the operational taxonomic unit (OTU) level, (**E**) PCoA applied the BrayCurtis distance metric at the OTU level for mice in the ND, HFD, and HFD + *K. latis* groups. (**F**–**H**) Principal Coordinate Analysis (PCoA) utilizing the Bray–Curtis distance was performed on mice at the Operational Taxonomic Unit (OTU) level in different diet groups, including normal diet (ND), high-fat diet (HFD), HFD with LP8198 supplementation, and HFD with LP8198 + *K. latis* supplementation. The data is presented as the average plus or minus the standard deviation, with a sample size of 6.

**Figure 8 foods-13-01124-f008:**
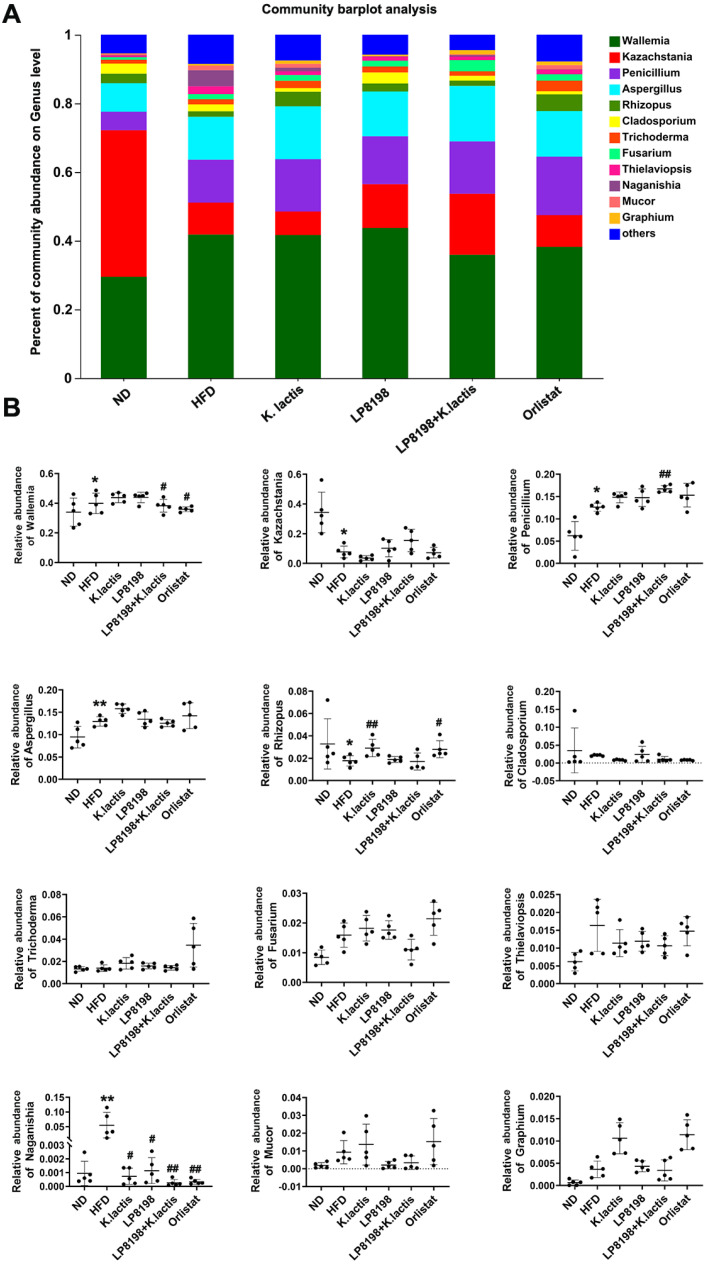
Examining how high-fat diet impacts the diversity of microorganisms in the colon at the genus level. (**A**) Taxonomic analysis of fungi at the genus level. (**B**) Genus-level distribution of gut microbiota relative abundance. The data is presented as the average plus or minus the standard deviation (*n* = 5). Statistical significance was evaluated using a one-way ANOVA followed by a Dunnett’s post hoc test for (**B**). * *p* < 0.05, ** *p* < 0.01 compared to the ND group; ^#^
*p* < 0.05, and ^##^
*p* < 0.01 compared to the HFD group.

**Table 1 foods-13-01124-t001:** Primers used for quantitative real-time PCR.

GenBank Accession Number	Gene	Sequences (5′-3′)	Melting Temperature	Amplicon Size
NM_007393.5	Actb (β-Actin)	Forward CTGAACCCTAAGGCCAACCG	59 °C	113
		Reverse CGACCAGAGGCATACAGGGA		
NM_011146.4	PPAR-γ	Forward GCAGGAGCAGAGCAAAGAGG	59 °C	196
		Reverse ATTCATCAGGGAGGCCAGCA		
NM_007988.3	Fasn	Forward CTACAGCATCGACGCCAGTC	58 °C	108
		Reverse TTCCACACCAGGCACAGGTA		
NM_133360.3	Acaca (ACC1)	Forward GCTGAGCTTCGGGGTGGTTC	58 °C	131
		Reverse CGGAATTTGATTTCTACTGT		
NM_033218.2	SREBP-2	Forward CCTCAAGTGCAAAGCCTCGT	60 °C	113
		Reverse AGTGTGCCATTGGCTGTCTG		
NM_013495.2	Cpt1a (CPT1)	Forward GTGACTGGTGGGAGGAATAC	56 °C	83
		Reverse GAGCATCTCCATGGC GTAG		
NM_011144.6	PPAR-α	ForwardAACATCGAGTGTCGAATATGTGG	60 °C	99
		Reverse CCGAATAGTTCGCCGAAAGAA		

## Data Availability

The original contributions presented in the study are included in the article/Appendix A, further inquiries can be directed to the corresponding author.

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
