# Peer review of "The Probiotic Kluyveromyces lactis JSA 18 Alleviates Obesity and Hyperlipidemia in High-Fat Diet C57BL/6J Mice"

_foods, 2024, doi:10.3390/foods13071124_

Round 1

Reviewer 1 Report

Comments and Suggestions for Authors

I reviewed the manuscript by the group of Ma on the utility of a yeast in improving metabolic health. Here are some suggestions to increase the publishability of this manuscript.

1. Please indicate the duration of the treatments in the methods?

2. Why was muscle actin (Acta1), which is poorly expressed in the liver, used as housekeeping gene instead of the more commonly used and more highly-expressed Actb?

3. The incorrect gene was used to create primers for fatty acid synthase (Fasn). NM_001146708 codes for Fas cell surface death receptor. 

4. Why was a partial cds used to design primers for ACC? Also, which ACC is being evaluated as there are 2 acetyl carboxylases in the mammals, Acaca and Acacb.

5. Why were short-chain fatty acid measured in the feces while fungal microbiota was assessed in the cecal contents?

6. There are some issues with the statistical analysis. Fig 1A should be analyzed using Repeated Measures ANOVA, not one way ANOVA. Why was Kruskall-Wallis performed on Fig 1B,F,H? Please standardized you posthoc tests.

7. Fig 1A,B - Weight loss was observed. If food intake is not unchanged, is the weight loss due to increased energy expenditure. Did the authors assess changes in the digestibility, energy absorption by assessing energy loss in the feces?

8. In Fig 1E - Fasting glucose was changed. Is this due to increased insulin sensitivity or fasting insulin?

9. Since the authors focus on the hepatic lipid regulation, can the author quantify hepatic lipids?

10. Can the authors provide a quantification of liver pathologies and adipocyte morphology?

11. Figure 4. There are inconsistencies with the figures and text. For example, Line 382 mentioned that isobutyrate, valeric, and isovaleric were decreased in HFD but the figures lack the significance marks. Please correct.

12. Is there an explanation why K. lactis had less effect on microbiota diversity compared to LP8198 as shown in Fig 5E vs Fig 5F 

13. Figure 6. Line 432-433 reads that high fat diet changes the distribution but when we look at Fig 6B, the relative abundance of major bacterial groups in NC and HFD are similar.

14. Similar to Figure 6, in Figure 8, the lack of statistical marks on the figure is confusing when the authors mentioned that there are significant differences between groups.

15. I caution the authors to write the probiotics directly controls metabolism. The data here are association studies. Energy balance control of complex and only gain or loss of function studies 

can ultimately decide the role of these microbes in regulating body weight and lipid metabolism. Thus, the authors should refrain from writing, for example, in Line 538-540, "This suggests that both LP8198 and K.lactis can control lipid metabolism by reducing adipogenesis and lipogenesis and elevating fatty acid oxida-tion, thereby causing fat loss and weight loss. Improvements in lipid profile can be just a consequence of weight loss whose mechanisms were not fully studied here.

16. If supplementing these microbes do not change the composition of the mouse microbiota, then what could be a proposed mechanism/s on how LP8198 and K.lactis reduce body weight and improve lipids?

17. Can the authors comment on the lack of synergistic effects when K. lactis and LP8198 was combined?

Minor:

1. Please correct the spelling of Kluveromces to Kluyveromyces.

2. In line 4, please correct bal-ance to balance and fugal to fungal.

3. In line 23, please correct triglycer-ide to triglyceride

4. In line 24, please correct inflamma-tory to inflammatory

5. Line 26, bacte-ria to bacteria

6. Line 27, mycobio-ta to mycobiota

7. There are many incorrectly hypenated words. Please go over the manuscript and correct them.

8. Line 56-59. The studies cited are all done in experimental model of metabolic disease. Please write these statement as this sounded like L. plantarum is routinely used to treat HUMAN metabolic disease.

9. Make sure the scientific names are italicized. Some of them are not.

10. Primer Table. Please correct the accession number and name of CPT to CPT1a

11. Please use the correct gene notations in the primer table. For example, actin should be written as Acta1

12. Line 211. Is this correct that "DNA extraction from soil samples"...? If incorrect, what type of samples were used for the microbiome DNA sequencing analysis?

13. Methods indicated that there are 10 mice per group (Line 110). Why there are only 6 replicates in Figure 1-4, n=5 in Figure 5

14. Can the bar graphs in Fig2,3 be converted to scatter plots similar to Fig 1 and 4?

15. Can the authors include the word "systemic" inflammation for results pertaining to Fig 2. It sounds like serum Tnf-a and Il1-b are measured only in the liver.

16. In line 369, the treatments did not affect markers of lipid oxidation so writing the K. lactis and LP8198 influence the activity of genes for lipid breakdown is not supported by data.

17. I suggest that the authors provide the statistical test that they use for each data on the figure legends.

18. I advise the authors not to write sentences like "There is a slight decrease or increase" when the there is no statistical difference at all. 

Reviewer 2 Report

Comments and Suggestions for Authors

The manuscript, “The probiotic Kluveromces lactis JSA 18 alleviates obesity and hyperlipidemia in high-fat diet C57BL/6J mice,” refers to using this strain as a probiotic capable of controlling mice's body weight.

The manuscript covers the interests of the Foods journal; however, it requires several improvements. The manuscript contains several typos and double spaces that need the Author’s attention. 

Here are some punctual suggestions:

2. Materials and methods: 

-              2.2 Test for tolerance to acid and bile. Please check the text, changing all forms, such as degrees Celsius and grams per litre, with the respective contract form (e.i., °C; g/l).

-              2.3 The strain name should be written in italics. 

-              2.7. Please specify what kind of cells were used for RNA extraction. Please number the tables. The authors should include in Table 1 some columns to report the amplicon size and the melting temperature of each PCR product, as well as the bibliographic reference referred to for each primer set. If they designed the primes, please specify the procedures adopted in a new section.

-              Please include a new section describing the analysis of the raw sequences.

3. Results.

Figure 1a is of low quality; please improve its quality. Please include the body weight at the beginning of the experiment as a first point of Figure 1A. This point allows the reader to track the evolution over time of the body weight of the different mouse groups.

Figures S2 and S3 should be moved in the text and not as supplementary data.

Round 2

Reviewer 1 Report

Comments and Suggestions for Authors

All of my concerns have been addressed.